# Recurrence Kinetics after Laparoscopic Versus Open Surgery in Colon Cancer. A Meta-Analysis

**DOI:** 10.3390/jcm10184163

**Published:** 2021-09-15

**Authors:** Ross Lilley, Evangeline Chan, Nicklaus Ng, Amber Orr, Marcin Szostok, Gloria Ting Ting Yeh, Ross Tulloch, George Ramsay, Zhirajr Mokini, Patrice Forget

**Affiliations:** 1Institute of Applied Health Sciences, University of Aberdeen, Aberdeen AB25 2ZD, UK; ross.lilley.17@abdn.ac.uk (R.L.); evangeline.chan.17@abdn.ac.uk (E.C.); nicklaus.ng.17@abdn.ac.uk (N.N.); amber.orr.17@abdn.ac.uk (A.O.); marcin.szostok.17@abdn.ac.uk (M.S.); t.t.yeh.17@abdn.ac.uk (G.T.T.Y.); ross.tulloch.17@abdn.ac.uk (R.T.); 2Rowett Institute of Nutrition and Health, University of Aberdeen, Aberdeen AB25 2ZD, UK; george.ramsay@abdn.ac.uk; 3Department of General Surgery, NHS Grampian, Aberdeen AB25 2ZN, UK; 4Independent Researcher, supported by the European Society of Anaesthesiology and Intensive Care Mentorship Programme, B-1000 Brussels, Belgium; zhirajrmokini@yahoo.com; 5Epidemiology Group, Institute of Applied Health Sciences, School of Medicine, Medical Sciences and Nutrition, University of Aberdeen, Aberdeen AB25 2ZD, UK; 6Department of Anaesthesia, NHS Grampian, Aberdeen AB25 2ZN, UK

**Keywords:** recurrence kinetics, laparoscopic surgery, open surgery, colon cancer, meta-analysis

## Abstract

Background: Colorectal cancer (CRC) is a leading cause of mortality worldwide and in the UK. Surgical resection is the main curative treatment modality available and using a laparoscopic vs. an open approach may have a direct influence on the inflammatory response, influencing cancer biology and potentially the recurrence kinetics by promoting cancer growth. Methods: This systematic review aims to compare laparoscopic with open surgery for the treatment of colon cancer with a specific focus on the moment of the recurrence. We included randomised controlled trials in intended curative surgery for colon cancer in adults. Interventions: Studies investigating laparoscopic vs. open resection as an intended curative treatment for patients with confirmed carcinoma of the colon. The two co-primary outcomes were the time to recurrence and the overall survival (OS) and disease-free survival (DFS) at three and five years. Meta-analyses were done on the mean differences. Results: After selection, we reviewed ten randomised controlled trials. Most of the trials did not display a statistically significant difference in either DFS or OS at three or at five years when comparing laparoscopic to open surgery. Groups did not differ for the OS and DFS, especially regarding the time needed to observe the median recurrence rate. The quality of evidence (GRADE) was moderate to very low. Conclusion: We observed no difference in the recurrence kinetics, OS or DFS at three or five years when comparing laparoscopic to open surgery in colon cancer.

## 1. Introduction

Colorectal cancer (CRC) is defined as cancer arising from the epithelium of the colon or rectum [1]. According to Cancer Research UK [2], it is the 4th most common cancer in the UK, constituting 11% of all new cancer diagnoses. It is also the second most common cause of cancer mortality in the UK. Survival is linked to stage at presentation: with early disease detection associated with better outcomes [3].

CRC is thought to have a multifactorial aetiology, with genetic factors, environmental exposures (diet, smoking and alcohol intake) and inflammatory conditions being implemented in its development [4]. Among the inflammatory condition, surgery is a scheduled one, and the reaction of tissue injury might be dependent on modifiable factors, like the surgical approach. Indeed, surgery is the most common treatment offered and sometimes considered to be the only curative modality [4]. Resection is often performed to remove the primary tumour [5] and can be done using different techniques, ranging from hemicolectomy, where one half of the colon is removed, to total colectomy, where the entire colon is removed [6]. Resection can be performed using different approaches, namely open or through a minimally invasive technique [7]. The open approach is the conventional method, but it is associated with more postoperative pain, a longer hospital stay and, importantly, a bigger inflammatory response which could, in turn, influence negatively the recurrence kinetics by promoting cancer growth [8]. In this context, separating the colon from rectal surgery is relevant. The disease is not only different in the rectum (treated with different neoadjuvant therapies for colon cancer), but the surgical approaches are also different. Laparoscopic hemicolectomy is a very different procedure from previous laparoscopic resections, but also includes specific challenges (and their inflammatory complications) with the rectum being in the pelvis.

### Aims and Objectives

Many systematic reviews do not distinguish between colon and rectal cancer, with little specific data on colon cancer. This paper aims to conduct a systematic review with meta-analysis, using available literature on randomised controlled trials (RCTs) across multiple databases, to compare the effectiveness of laparoscopic versus open surgery for the intended curative resection of colon cancer in adults. The primary outcomes analysed are overall survival (OS) and disease-free survival (DFS) at both three and five years. The secondary objective is the comparison of the time to median recurrence rate.

## 2. Materials and Methods

### 2.1. Inclusion and Exclusion Criteria

We established the inclusion and exclusion criteria using the PICOS tool [9], which aided to focus our question. Table 1 summarises the research criteria.

### 2.2. Search Strategy

We conducted literature search on Ovid MedLine, Ovid Embase, Cochrane Central Register of Controlled Trials and PubMed. Appendix A reflects the search terms, combinations and applied limits used.

Two investigators assessed independently each database and compared the screening. This helped to ensure consistency and eradicate possible bias. After initial screening and removal of duplicates we compiled a final list of publications for review, as displayed in Figure 1. This search methodology is based on the PRISMA (Preferred Reporting Items of Systematic reviews and Meta-Analyses) flowchart.

### 2.3. Data Collection and Analysis

Two independent investigators extracted data from selected articles [11,12,13,14,15,16,17,18,19,20] and the research group verified it by using an independently created data extraction form (Appendix A). In one case we found a second publication from the same trial concerning colorectal cancer being published including only data for left colon cancer. When it was not possible to obtain the raw data from the authors, we extracted data from the figures using GetData Digitizer software V 2.26.0.20 (GetData Graph digitizer software version 2.25) assuming a normal distribution for the data [21] to perform the meta-analysis. We verified the accuracy and the precision of the tool across the graphs used and found the worst result of +/−0.69% in Tung et al. [11].

We performed a group discussion to resolve discrepancies and compile data. To determine the quality of evidence extracted from all included studies for which *p*-values were available we used the GRADE approach [22]. As all studies were RCTs, the evidence in all cases was initially deemed ‘high quality’. For serious limitations in study design or risk of bias we downgraded the quality of evidence by one level (or by two for very serious). In case of unexplained inconsistency, indirectness of evidence, imprecision of results or high probability of publication bias, we downgraded evidence by a further level [23].

We compared the mean survival and disease-free periods between groups. We computed quartiles and calculated the interquartile range. The interquartile range (IQR) divided by 1.35 is a robust measure that describes statistic dispersion and can serve as a surrogate of standard deviation [24].

We performed a meta-analysis for comparison of means for continuous data using RevMan V. 5.4.1 software (Cochrance Collaboration, Copenhagen, Denmark). Given the heterogeneity of studies we used the random effect method.

### 2.4. Critical Appraisal

Two independent reviewers performed a critical appraisal for each paper, using a checklist developed by Scottish Intercollegiate Guidelines Network (SIGN) [25], to evaluate the validity of RCTs and their results. All members performed an appraisal at least twice. Appendix A shows the characteristics of each study.

## 3. Results

### 3.1. Study Characteristics

The literature search produced 1990 articles. Following title and abstract screening and full-text review, with inclusion and exclusion criteria applied, we identified a total of ten RCTs [11,12,13,14,15,16,17,18,19,20] for inclusion. No trial carried out any blinding procedure due to the nature of the surgical procedures. Three trials were multi-centre [15,16,19], and seven single-centre [11,12,13,14,17,18,20]. Analysis involved in total 3610 colon cancer patients. Total of 1808 patients underwent laparoscopic surgery and 1802 underwent conventional open surgery for intended curative resection of a primary tumour. Three trials [15,16,19] assessed the non-inferiority of laparoscopic compared to open resection, the remaining seven trials [11,12,13,14,17,18,20] aimed to assess superiority. Almost all studies included data on both DFS and OS; the exception being Chung et al. [14] that measured only OS.

### 3.2. Population

All selected studies investigated laparoscopic vs. open resection as an intended curative treatment for patients with confirmed carcinoma of the colon. The age ranges for participants included: no limit [18], ≥18 years [11,12,13,14,15,19], ≥20 years [16,20] and ≥75 years [17]. Table 2 summarises the differences in tumour location. Stage of colon cancer differed between individuals in all studies and ranged from T0 to T4. However, separation of outcomes for each stage was beyond the scope of this review. Recurrent exclusion factors that we included were metastatic disease, pregnant or lactating women and patients considered unfit for surgery. The conduction of RCTs involved many countries, including Hong Kong, Japan, the USA and several European countries. All studies involved surgeons with adequate training to perform laparoscopic surgery.

### 3.3. Intervention

All included studies in this review assessed the use of non-robotic, laparoscopic resection of the colon with curative intent, with Chung et al. [14] using a hand-assisted technique. The only study to use a co-intervention was Tung et al., [11] who assessed the efficacy of endoscopically inserted, metal stents prior to laparoscopic resection of cancer obstructing the left colon. Nine studies [11,12,13,14,15,16,17,19,20] reported the use of adjuvant chemotherapy in a proportion of participants. The chemotherapy agent varied between studies and included oral 5-fluorouracil derivatives, [16,17] leucovorin, [16] FOLFOX regimen [12,13], fluoropyrimidine [20] and un-specified regimens [11,14,15]. The proportion of patients receiving chemotherapy was balanced in intervention and comparator groups [19] in all studies. Three trials [15,16,20] specified that adjuvant chemotherapy was administered in stage III patients. Two studies [14,18] reported only on right laparoscopic hemicolectomy and one [13] reported only on left laparoscopic hemicolectomy. The remainder of the studies [11,15,16,17,19,20] included patients with varying tumour locations, which in turn governed the resection site. Toritani et al. [20] was the only study to report the use of complete mesocolic excision. Lymph node harvesting occurred during laparoscopic surgery in all studies.

### 3.4. Comparator

Conventional open resection of colon cancer with curative intent acted as a control in all studies. The only study to compare a laparoscopic intervention to emergency laparotomy was Tung et al. [11] in colonic obstruction. As with the laparoscopic group, two studies [14,18] reported on right hemicolectomy and one study [13] reported on left hemicolectomy in control groups. In the other six studies [11,15,16,17,19,20], the section of colon resected reflected tumour location. Toritani et al. [20] reported the use of complete mesocolic excision in both groups. Lymph node harvesting was performed during open surgery in all studies.

### 3.5. Outcomes

#### 3.5.1. Overall Survival

All studies provided data for OS (Table 3). Two studies listed OS as their primary outcome [11,16], and eight considered it as a secondary outcome [12,13,14,15,17,18,19,20]. The trials reported results at various follow-up times. Braga et al. [12,13] only published 5-year survival results. Conversely, Ishibe et al. [17] only provided 3-year survival results. Six studies [11,14,15,16,18,20] reported 5-year OS. The COLOR trial [19] was the only study to state both 3- and 5-year OS clearly in their paper. After a follow-up of three years, two studies [17,19] reported OS which was not different (i.e., *p* > 0.05). At 5-year follow-up, six studies [11,13,14,15,18,20] did not find any statistical significance in OS between the laparoscopic and open resection groups. Meta-analysis revealed no difference between groups in the survival periods (Figure 2).

#### 3.5.2. Disease-Free Survival

DFS was a primary outcome in three studies [11,15,19] and as a secondary outcome in seven (Table 4) [12,13,14,16,17,18,20]. Chung et al. [14] did not provide any data on this outcome. Follow-up periods for the reported values varied between studies. Braga et al. [12,13] provided no data on 3-year DFS. Conversely, Ishibe et al. [17] provided no data regarding 5-year DFS. Five other studies [11,15,16,18,20] reported 5-year DFS in their report without explicitly stating data on 3-year DFS; in these five studies, we extracted manually the values for 3-year DFS from Kaplan–Meier curves. Only the COLOR trial [19] stated the values for both 3- and 5-year DFS directly in their report. Ishibe et al. [17] was the only study to find no statistically significant difference in 3-year DFS between the two arms of their study. The COLOR trial [19] suggested that laparoscopic surgery was clinically non-inferior to open colectomy. At 5-years follow-up, six studies [11,12,13,15,18,20] found no significant difference between the laparoscopic and open groups. The results of two of these studies, Braga et al. [13] and Li et al. [17] excluded patients with stage IV disease. Meta-analysis revealed no difference between groups in the disease-free periods (Figure 3).

#### 3.5.3. Meta-Analysis

Studies presented a high within-studies variance. As shown in Figure 2 and Figure 3, at least 70% of the observed variance between studies is due to real differences in the effect size whereas less than 30% of the observed variance would have been expected based on random error.

### 3.6. Bias Assessment

#### 3.6.1. Selection Bias

All studies involved in this review used randomisation to limit baseline differences between the patient groups. (Appendix A) Six [12,13,14,15,18,19,20] used computer-generated lists to reduce the risk of selection bias. The remaining four [14,15,16,17] did not detail how lists were generated. Five [15,16,17,19,20] studies employed stratification to minimise imbalance between the size of the groups. Studies used tumour location or the proposed resection method as predominant variables. Braga et al., Chung et al., and Li et al., [12,13,14,18] noted the use of independent persons to deliver allocation information via sealed envelopes prior to surgery carries risks of bias [23]. The remaining six studies did not describe allocation concealment in sufficient detail for comment.

#### 3.6.2. Performance Bias

Due to the surgical nature of the interventions, all studies had considerable performance bias, hence a high risk for overall bias. Further, five studies [12,13,16,17,20] noted adjuvant chemotherapy provision for some participants.

#### 3.6.3. Detection Bias

None of the studies provided information on the assessors used in follow-up. However, failure to pick up recurrence might have affected DFS.

#### 3.6.4. Attrition Bias

Two studies did not report attrition [14,20]. Five of the remaining papers [12,13,15,17,18,19] reported low dropout rates in their studies of 5% or less. Two studies [11,16] reported dropout rates to be higher than 5%—the JCGOG0404 trial, 16 which reported attrition of 6% and Tung et al., [11] which reported attrition of 27%. Attrition rates of 20% or higher represent a high risk of attrition bias, and rates between 5% and 20% have a small risk of bias. [26] Therefore, there is a risk of attrition bias present in these two papers, although the risk of bias in the JCGOG0404 trial is low. Common reasons for attrition were metastatic disease at surgery leading to exclusion from the study and patients lost to follow-up.

#### 3.6.5. Reporting Bias

No paper reported any conflicts of interest. Seven reported no external funding. The COST trial received a number of grants from the National Cancer Institute [15] and the COLOR trial received funding from Ethicon Endo-surgery and the Swedish Cancer Foundation [19]. Neither corporation influenced initiation, design or any other aspect of the study.

## 4. Discussion

We evaluated OS, DFS and recurrence kinetics after laparoscopic vs. open surgical approach for colon cancer resection. There is probably no difference in these outcomes (moderate to very low level of certainty). As there is uncertain evidence about the differences in the inflammatory response after laparoscopy vs. laparotomy for colon cancer, we cannot formally conclude on the absence of effect of inflammatory processes on the outcome. However, we can reasonably conclude that the effect of the surgical approach on the inflammatory response, if any, is probably not sufficient to influence these outcomes.

Increased inflammation or immunosuppression caused by surgical technique may favour cancer proliferation after surgery, depending on the type of inflammatory mediators produced [27,28]. However most studies come from often small laboratory experiments or trials, while what happens in the human body is incredibly more complex [29]. Here, we evaluated OS and DFS after laparoscopic vs. open approach for colon cancer resection.

All included papers discussed these outcomes. Four trials [11,15,16,19] had either- or both outcomes as their primary endpoint, and six [12,13,14,17,18,20] had them as part of their secondary endpoint. Both outcomes were further split into 3-year OS/DFS, and 5-year OS/DFS.

Due to colon cancer’s potential to progress into a higher-grade cancer, it was difficult for the patients recruited to have open surgery or laparoscopic surgery as their sole intervention throughout the study. The JCOG0404 trial [16] randomly assigned 1057 patients at the beginning to two treatment arms: laparoscopic resection and open resection. However, of the 520 that had open surgery, 174 (33.5%) had adjuvant chemotherapy, and of the 525 that had laparoscopic surgery, 199 (37.9%) underwent adjuvant chemotherapy. These patients, however, were included with patients that did not undergo this additional treatment. Thus, as the adjuvant chemotherapy could have been a confounding factor, data for survival outcomes may have been affected. This confounder was also present in the trial by Braga et al., [12] where adjuvant chemotherapy was given to 118 (62.1%) patients in the laparoscopic surgery arm, and 124 (61.6%) patients in the open surgery arm, the COLOR trial [19] and Toritani et al. [20]. In addition, the majority of papers [12,13,16,17,20] did not report on the chemotherapy regime or length of treatment. No papers mentioned the dosage of chemotherapy given to the patients. These factors may have greatly affected outcomes but also open a new avenue of research on the need to consider the perioperative management (and treatments) of cancer as a whole in patients (before, during and after surgery).

There were some inconsistencies regarding patient cohort characteristics across the trials. Ishibe et al. [17] recruited patients >75 years of age, Toritani et al. [20] recruited patients >20 years of age, the JCOG0404 trial [16] recruited patients from 20–75 years old, Tung et al. [11] did not mention the age group of their patients, and the other trials recruited patients >18 years. As Ishibe et al. [17] had produced data from a patient-population of the elderly, it may be inappropriate to compare this set of data to data from other trials studying patients of a wider age range. The presence of co-morbidities is more likely among elderly patients, which will influence the data on OS and DFS rates. Age, and potentially frailty, is particularly relevant to consider in the perioperative context but insufficient, and the limited sample size here precludes any specific conclusion on that [30,31,32,33].

### 4.1. Quality of Evidence (GRADE)

The quality of evidence ranged from moderate to very low across the studies included in this review. All studies, excluding Li et al., [18] were downgraded by at least one level as they were deemed high risk of bias according to the Cochrane RoB 2.0 tool (Cochrane Collaboration, Copenhagen, Denmark) [34]. This was mainly due to a lack of blinding in the surgical setting of the trials. Ishibe et al. [17] and Toritani et al. [20] were marked down by a further level due to risk of bias in the measurement of outcomes. The quality of evidence from Ishibe et al. [17] and Tung et al. [11] was further lowered due to indirectness of the population (elderly participants) and intervention (co-intervention with endoscopic stenting) respectively. Imprecision due to a lack of power towards survival outcomes lowered the quality of evidence in all six superiority trials and the JCOG0404 trial [16]. In the superiority trials, this was due to assessment of these outcomes as secondary outcomes. The JCOG0404 trial [16] authors aimed to power their trial to 5-year OS, but they failed to accrue an adequate number of participants to reach their intended power.

The meta-analysis carried out in this review included RCTs in the present systematic review, in particular, the COLOR trial [19]. Although this trial could not statistically prove non-inferiority, when combined with other studies in meta-analysis, no statistically significant difference between laparoscopic surgery and open surgery was seen. We recommend an updated meta-analysis looking at 3- and 5-year survival so that an assessment of effect sizes can be carried out with the inclusion of studies published since Di et al. [35].

### 4.2. Limitations

As mentioned, the quality of multiple studies used in this review were downgraded due to lack of blinding, concerns about outcome measurement, and indirectness of population and intervention. This brings to question the validity of the data analysed, as some may be inaccurate.

Data that were not directly reported in some studies, such as 3-year survival, were extracted from Kaplan–Meier graphs. The limitation of this is that p-values have not been calculated. However, it is reasonable to extrapolate 5-year confidence intervals to 3-year data, which would make these data more robust.

There were several studies in which data might possibly be unreliable due to its small sample size. Chung et al., [14] Tung et al., [11] and Toritani et al. [20] reported that the number of patients they had recruited and randomised into the trial was 81, 48 and 66 respectively. Their sample sizes are significantly smaller than other trials analysed in this systematic review (COLOR trial [19]—1248 patients, and JCOG040 trial [16]—1057 patients). Trials with small sample sizes risk false-positive results and over-estimation of differences in outcomes [36,37,38,39]. In this case, any difference in outcomes between the laparoscopic and open surgery groups might appear to be overstated. Overall, this warrants further studies to reduce uncertainty. Other future human studies could include, for example, better staging of colon cancer patients allowing for more accurate stratification.

## 5. Conclusions

This meta-analysis shows that laparoscopic surgery appears to have a similar outcome, and specifically similar recurrence dynamics, when compared to open surgery for the intended curative treatment of colon cancer in the adult population, in relations to OS and DFS at both three and five years. Future research is needed on other cancers or in patients with different profiles (metastatic diseases, specific conditions like advanced age or frailty) to study whether the same conclusion can be reached. Finally, we suggest considering as a whole the perioperative care and treatments of the cancer patients.

## Figures and Tables

**Figure 1 jcm-10-04163-f001:**
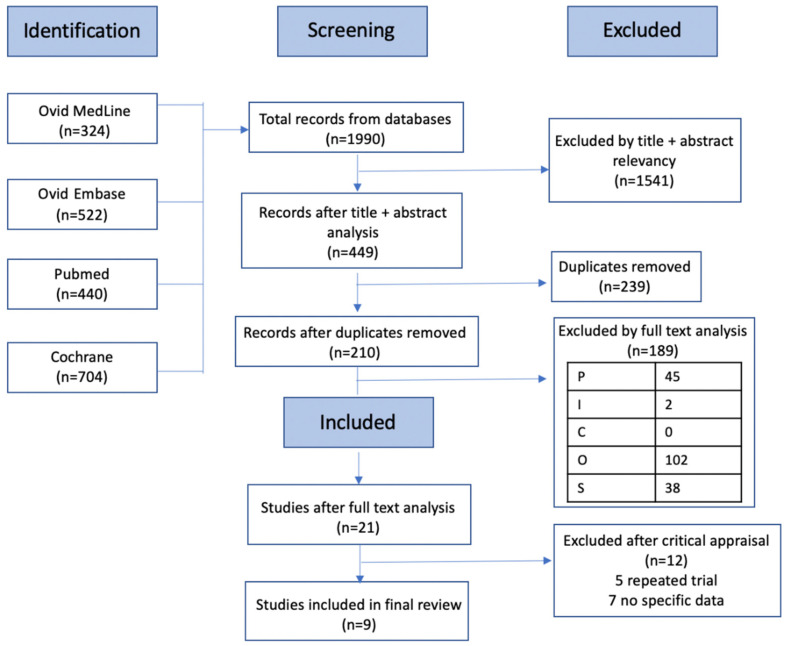
Flow diagram displaying literature search protocol. Adapted from PRISMA 2009 flow diagram [10].

**Figure 2 jcm-10-04163-f002:**
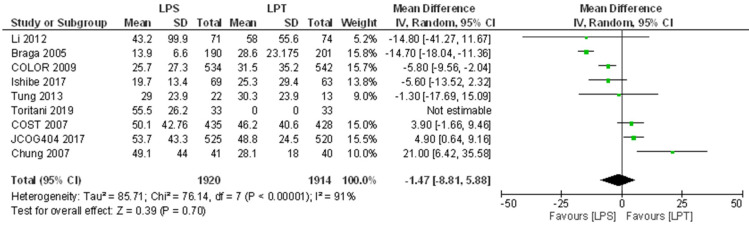
Forest plot of comparison: Mean difference for the overall survival. LPS: laparoscopic approach; LPT: open approach (laparotomy).

**Figure 3 jcm-10-04163-f003:**
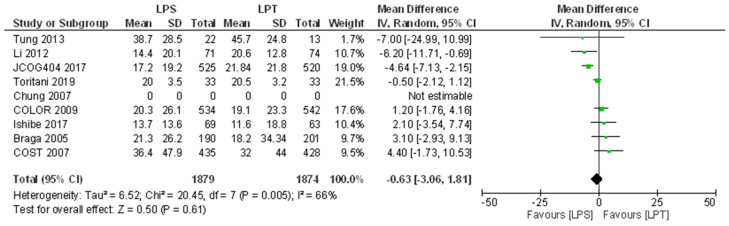
Forest plot of comparison: Mean difference for the disease-free survival. LPS: laparoscopic approach; LPT: open approach (laparotomy).

**Table 1 jcm-10-04163-t001:** Inclusion and exclusion criteria used in the literature search.

**Patients**	Adult population >18 years Colon cancer (all stages)	Paediatrics <18 years Metastatic disease Rectal cancer
**Intervention**	Intended curative laparoscopic surgery of primary tumour	
**Control**	Intended curative open surgery of primary tumour	
**Outcomes**	3-year disease-free survival 3-year overall survival 5-year disease-free survival 5-year overall survival Survival period Disease-free period	
**Study Design**	Randomised controlled trials published in English	All other study types

**Table 2 jcm-10-04163-t002:** Summary of population characteristics. The only data available are as follows and apply to both colon and rectal cancer patients: in the laparoscopic surgery group, 54 patients were <80 years and 44 were ≥80 years. For open surgery, 53 were <80 years and 39 were ≥80 years.

Study	Intervention	Control	Population	Average Age (Year)		Sample Size Analysed		Male Sex (%)		
				Laparoscopic	Open Resection	Laparoscopic	Open Resection	Laparoscopic	Open Resection	95% CI
Braga et al., 2005	Laparoscopic surgery	Open surgery	≥18 years Colorectal cancer	65 (13) mean (SD)	67 (11) mean (SD)	190	201	60	60	−14.70 [−18.04,−11.36]
Braga et al., 2010	Laparoscopic surgery	Open surgery	≥18 years Carcinoma of left colon	62.9 ^†^	64.9 ^†^	78	89	51 ^†^	53 ^†^	
Chung et al., 2007	Hand-assisted laparoscopic colectomy	Open colectomy	≥18 years Carcinoma of cecum or ascending colon	71	72.5	41	40	61	65	21.00 [6.42,35.58]
COLOR, 2009	Laparoscopic surgery	Open resection	≥18 years Carcinoma of caecum, ascending colon, descending colon or sigmoid colon	71	71	534	542	52	53	−5.80 [−9.56,−2.04]
COST, 2007	Laparoscopically assisted colectomy	Open colectomy	≥18 years Carcinoma of right, left or sigmoid colon	70	69	435	428	51	49	3.90 [−1.66,9.46]
Ishibe et al., 2017	Minimally invasive laparoscopic resection	Conventional open resection	≥75 years Adenocarcinoma of colon and rectum	NI	NI	69	63	50	60	−5.60 [−13.52,2.32]
JCOG404, 2017	Laparoscopic surgery	Open surgery	20–75 years Carcinoma of caecum, ascending colon, sigmoid colon	64	64	525	520	54	60	4.90 [0.64,9.16]
Li et al., 2012	Laparoscopic assisted right hemicolectomy	Open right hemicolectomy	All ages ‡ carcinomas of caecum, ascending colon, hepatic flexure or transverse colon	68	68	71	74	46	43	−14.80 [−41.27,11.67]
Toritani et al., 2019	Laparoscopic surgery	Conventional open surgery	≥20 years Transverse and descending colon cancer	64	67	33	33	73	48	
Tung et al., 2013	Endolaparoscopic resection	Conventional open surgery	≥18 years Obstructed left sided colon cancer.	64.5	68.5	22	13	58	50	−1.30 [−17.69,15.09]

Data for colon and rectal cancer combined, as paper did not separate these two groups. † Data for cancer and non-cancer patients combined, as paper did non separate these two groups. ‡ Authors were contacted to confirm that all the participants were over 18 years old in accordance with the inclusion criteria of this review, refer to Appendix A.

**Table 3 jcm-10-04163-t003:** Summary of overall survival at 3 and 5 years and quality of evidence. Data extracted from Kaplan–Meier curves using GetData Graph Digitizer. Excluding patients with stage IV disease.

Overall Survival	3-Year Follow-Up (%)			5-Year Follow-Up (%)			LPS			LPT						
Study	LPS	LPT	*p*	LPS	LPT	*p*	Mean	SD	Patients	Mean	SD	Patients	Weight	95% CI	Quality of Evidence (GRADE)	Comments
Li 2012	83.4	86.1	NA	74.2	75.0	0.835	43.2	99.9	71	58.0	55.6	74	5.2%	−14.80 [−41.27,11.67]	⊕⊕⊝⊝ Low ^a,e^	There may have been no difference in 5-year OS between laparoscopic and open surgery.
Braga 2005	NI	NI	NA	72	66	0.321	13.9	6.6	190	28.6	23.175	201	15.8%	−14.70 [−18.04,−11.36]	⊕⊕⊝⊝ Low ^a,e^	There may have been no difference in 5-year OS between laparoscopic and open surgery.
Braga 2010	NI	NI	NA	61.1	56.5	0.16–0.65 according to stage	NA	NA	NA	NA	NA	NA	0	Not estimable	⊕⊕⊝⊝ Low ^a,e^	There may have been no difference in 5-year OS between laparoscopic and open surgery.
COLOR 2009	81.8	84.2	0.45	73.8	74.2	NI	25.7	27.3	534	31.5	35.2	542	15.7%	−5.80 [−9.56,−2.04]	⊕⊕⊕⊝ Moderate ^a^	There is probably no difference in 3-year OS after laparoscopic surgery compared to open surgery.
Ishibe 2017	93.9	93.5	0.901	NI	NI	NA	19.7	13.4	69	25.3	29.4	63	13.8%	−5.60 [−13.52,2.32]	⊕⊕⊕⊝ Moderate ^a^	There is probably no difference in 5-year OS after laparoscopic surgery compared to open surgery.
Tung 2013	71	46	NA	48	27	0.076	29.0	23.9	22	30.3	23.9	13	9.0%	−1.30 [−17.69,15.09]	⊕⊝⊝⊝ Very Low ^b,d,e^	There may have been no difference in 3-year OS after laparoscopic compared to open surgery, but the evidence is uncertain.
Toritani 2019	97.1	100.0	NA	93.3	100.0	0.543	55.5	26.2	33	0.0	0.0	33	0	Not estimable	⊕⊕⊝⊝ Low ^a,e^	5-year OS after Laparoscopic surgery may not have been non-inferior to open surgery.
COST 2007	86.8	86.8	NA	76.4	74.6	0.93	50.1	42.76	435	46.2	40.6	428	15.0%	3.90 [−1.66,9.46]	⊕⊕⊕⊝ Moderate ^e^	There is probably no difference in 5-year OS after laparoscopic surgery compared to open surgery.
JCOG404 2017	96	95.8	NA	91.8	90.4	0.073†	53.7	43.3	525	48.8	24.5	520	15.5%	4.90 [0.64,9.16]	⊕⊝⊝⊝ Very Low ^b,d,e^	There may have been no difference in 5-year OS after laparoscopic compared to open surgery, but the evidence is uncertain.
Chung 2007	95	86	NA	83	74	0.90	49.1	44.0	41	28.1	18.0	40	10.0%	21.00 [6.42,35.58]	⊕⊝⊝⊝ Very Low ^a,d,e^	There may have been no difference in 5-year OS after laparoscopic compared to open surgery, but the evidence is uncertain.

GRADE Working Group grades of evidence: ⊕⊕⊕⊕—High quality: further research is very unlikely to change our confidence in the estimate effect. ⊕⊕⊕⊝—Moderate quality: further research is likely to have an important impact on our confidence in the estimate of effects. ⊕⊕⊝⊝—Low quality: further research is very likely to have an important impact on our confidence in the estimate of effects. ⊕⊝⊝⊝—Very low quality: we are very uncertain about the estimate. LPS = laparoscopy, LPT = laparotomy. *p*-non-inferiority value. ^a^ Downgraded one level due to risk of bias or limitations in the detailed design and implementation. ^b^ Downgraded two levels due to risk of bias or limitations in the detailed design and implementation. ^c^ Downgraded one level due to unexplained heterogeneity or inconsistency of results. ^d^ Downgraded one level due to indirectness of evidence. ^e^ Downgraded one level due to imprecision of results. ^f^ Downgraded one level due to high probability of publication bias.

**Table 4 jcm-10-04163-t004:** Summary of disease-free survival at 3 and 5 years and quality of evidence. Data extracted from Kaplan–Meier curves using GetData Graph Digitizer. Excluding patients with stage IV disease.

Disease-Free Survival	3-Year Follow-Up (%)			5-Year Follow-Up (%)			LPS			LPT						
Study	LPS	LPT	*p*	LPS	LPT	*p*	Mean	SD	Patients	Mean	SD	Patients	Weight	95% CI	Quality of Evidence (GRADE)	Comments
Tung 2013	77	78	NA	52	48	0.63	38.7	28.5	22	45.7	24.8	13	1.7%	−7.00 [−24.99,10.99]	⊕⊝⊝⊝ Very Low ^a,d,e^	There may have been no difference in 5-year DFS after laparoscopic compared to open surgery but the evidence is uncertain.
Li 2012	84.3	86.3	NA	82.3	84.1	0.78	14.4	20.1	71	20.6	12.8	74	10.7%	−6.20 [−11.71,−0.69]	⊕⊕⊕⊝ Moderate ^e^	There is probably no difference in 5-year DFS after laparoscopic surgery compared to open surgery.
JCOG0404 2017	80.1	82	NA	80	79	NI	17.2	19.2	525	21.84	21.8	520	19.0%	−4.64 [−7.13,−2.15]	⊕⊕⊝⊝ Low ^a,e^	NA
Toritani 2019	90.5	87.2	NA	90.5	87.3	0.752	20.0	3.5	33	20.5	3.2	33	21.5%	−0.50 [−2.12,1.12]	⊕⊝⊝⊝ Very Low ^b,d,e^	There may have been no difference in 5-year DFS after laparoscopic compared to open surgery but the evidence is uncertain.
COLOR 2009	74.2	76.2	0.70, 0.030 ^†^	66.5	67.9	NI	20.3	26.1	534	19.1	23.3	542	17.6%	1.20 [−1.76,4.16]	⊕⊕⊕⊝ Moderate ^a^	3-year DFS after laparoscopic surgery is probably not non-inferior to open surgery.
Ishibe 2017	89.6	91.5	0.73	NI	NI	NA	13.7	13.6	69	11.6	18.8	63	10.4%	2.10 [−3.54,7.74]	⊕⊝⊝⊝ Very Low ^b,d,e^	There may have been no difference in 3-year DFS after laparoscopic compared to open surgery but the evidence is uncertain.
Braga 2005	NI	NI	NA	64.5	60.2	0.55–0.81 according to stage	21.3	26.2	190	18.2	34.34	201	9.7%	3.10 [−2.93,9.13]	⊕⊕⊝⊝ Low ^a,e^	There may have been no difference in 5-year DFS between laparoscopic and open surgery.
Braga 2010	NI	NI	NA	63	63	0.405	NA	NA	NA	NA	NA	NA	0	Not estimable	⊕⊕⊝⊝ Low ^a,e^	There may have been no difference in 5-year DFS between laparoscopic and open surgery.
COST 2007	80.4	79.2	NA	69.2	68.4	0.94	36.4	47.9	435	32.0	44.0	428	9.5%	4.40 [−1.73,10.53]	⊕⊕⊕⊝ Moderate ^a^	There is probably no difference in 5-year DFS after laparoscopic surgery compared to open surgery.
Chung 2007	NI	NI	NA	NI	NI	NA	NA	NA	NA	NA	NA	NA	0	Not estimable	⊕⊕⊝⊝ Low ^a,e^	N/A

GRADE Working Group grades of evidence: ⊕⊕⊕⊕—High quality: further research is very unlikely to change our confidence in the estimate effect. ⊕⊕⊕⊝—Moderate quality: further research is likely to have an important impact on our confidence in the estimate of effects. ⊕⊕⊝⊝—Low quality: further research is very likely to have an important impact on our confidence in the estimate of effects. ⊕⊝⊝⊝—Very low quality: we are very uncertain about the estimate. LPS = laparoscopy, LPT = laparotomy. ^†^
*p*-non-inferiority value. ^a^ Downgraded one level due to risk of bias or limitations in the detailed design and implementation. ^b^ Downgraded two levels due to risk of bias or limitations in the detailed design and implementation. ^c^ Downgraded one level due to unexplained heterogeneity or inconsistency of results. ^d^ Downgraded one level due to indirectness of evidence. ^e^ Downgraded one level due to imprecision of results. ^f^ Downgraded one level due to high probability of publication bias.

## Data Availability

All the data are included in the manuscript.

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
