# Peer review of "Recurrence Kinetics after Laparoscopic Versus Open Surgery in Colon Cancer. A Meta-Analysis"

_jcm, 2021, doi:10.3390/jcm10184163_

Round 1
Reviewer 1 Report
Well written paper, however my greatest concern is already mentioned in the discussion: the quality of evidence of randomised controlled trials that were analysed ranged from moderate to VERY low across the studies and we might ask ourselves, why would we like to do a meta-analysis on so low quality data?
Author Response
The reviewer is right when writing "the quality of evidence of randomised controlled trials that were analysed ranged from moderate to VERY low across the studies and we might ask ourselves, why would we like to do a meta-analysis on so low quality data?"
However, we believe this was not predictable. Eventually, it may justify further studies, and we have added the following sentence: "Overall, this warrants further studies to reduce uncertainty."
Reviewer 2 Report
The possible relationship between inflammatory insult in open surgery and colon cancer recurrene, is very interesting. These data, extrapolated from pre clinical studies, require further future human studies. In the current state of the art, the recurrence is more or less the same whether it is open surgery or laparoscopy. Howerer, the staging of colon cancer patients should be perfected soon after surgery, with further tests such as lymfoscintigraphy a test useful in the study of micrometastasis. Unfortunately, even in T1,2N0-M0 patients, the recurrence a 3/5 years has been demonstrated.
Your paper is of a good standard.
Author Response
We thank the reviewer for the interesting comment. Staging may in the future include inflammatory profile among other markers, although not directly evidenced by our work.
According to the reviewer's comment, we have added the following sentence in the discussion: "Other future human studies could include, for example, better staging of colon cancer patients allowing for more accurate stratification."